# Spatial distribution of three ARGONAUTEs regulates the anther phasiRNA pathway

Hinako Tamotsu[1,4], Koji Koizumi[2,4], Alejandro Villar Briones [3] & Reina Komiya [1] ✉

Argonaute protein (AGO) in association with small RNAs is the core machinery of RNA silencing, an essential mechanism for precise development and defense against pathogens in many organisms. Here, we identified two AGOs in rice anthers, AGO1b and AGO1d, that interact with phased small interfering RNAs (phasiRNAs) derived from numerous long non-coding RNAs. Moreover, 3D-immunoimaging and mutant analysis indicated that rice AGO1b and AGO1d cell type-specifically regulate anther development by acting as mobile carriers of these phasiRNAs from the somatic cell layers to the germ cells in anthers. Our study also highlights a new mode of reproductive RNA silencing via the specific nuclear and cytoplasmic localization of three AGOs, AGO1b, AGO1d, and MEL1, in rice pollen mother cells.

Site specificity of reproductive Argonaute proteins (AGOs) is essential for accurate germline development via transposable element silencing in many species[1-4]. MEIOSIS ARRESTED AT LEPTOTENE1 (MEL1) is a germ cell-specific AGO among 19 AGO family members in rice[5]. MEL1 interacts with 21-nucleotide (nt) phasiRNAs that have cytosine at the 5′-terminal position (C-phasiRNAs), which are derived from more than 770 long non-coding RNAs (lncRNAs) named *21PHAS*s during pre-meiosis and early meiosis in rice[6]. During plant reproduction, numerous 21- and 24-nt phasiRNAs are enriched in the anthers, the male organ in plants, via cleavage of miR2118/2275 and processing by Dicer-like proteins[7-12]. *Trans*-acting cleavage of 21-nt phasiRNAs in the cytoplasm of male germ cells has been detected[13,14]. In addition, 24-nt maize meiotic phasiRNAs and OsRDR6-dependent 24-nt rice small RNAs influence CHH DNA methylation during meiosis, leading to *cis* regulation by 24-nt phasiRNAs for maintaining normal meiosis in grasses[15,16]. Moreover, recent studies revealed that 24-nt siRNAs move from the tapetum to the meiocytes[17,18], implying non-cell-autonomous regulation via small RNAs in anthers. Since somatic anther wall development affects germ cell development, their synchronization in anther development is indicative of the importance of cell-to-cell interaction[19-21]. Intercellular communication via mobile signals is critical for cell fate determination in higher plants[22,23]. However, the non-cell-autonomous mechanism in anther development and the

meiotic functions of 21-nt phasiRNAs remain unknown, although MEL1 interaction with 21-nt C-phasiRNAs plays an important role in the pairing between homologous chromosomes during early meiosis in rice[6]. Therefore, focusing on reproductive AGOs as interactors of these phasiRNAs may illuminate the molecular roles of phasiRNAs in male organ development and in enhancing reproductive competence via the non-cell-autonomous developmental system in many species.

AGO1 generally binds to miRNA to execute post-transcriptional gene silencing in land plants[24]. There are four members (AGO1a–d) in the AGO1 subfamily in rice, and AGO1a, b, and c bind to miRNAs at vegetative stages[25], as does *Arabidopsis* AtAGO1[26]. Rice *AGO1b/c/d* expression increases specifically during reproduction, while *AGO1a* is enriched at the vegetative stage. Our recent proteome and mRNA localization analyses have identified AGO1b and AGO1d as candidate miR2118-dependent soma AGOs[27], suggesting different functions of rice AGO1 subfamily proteins between vegetative and reproductive stages.

In this study, we identified phasiRNAs loaded onto rice AGO1b and AGO1d through small RNA-immunoprecipitation (RIP) and examined the role of AGO1b and AGO1d in anther development. We also successfully performed three-dimensional (3D)-immunostaining using whole anthers, which enabled us to distinguish the cell types and to identify the specific subcellular localization of AGO1b/d in each

[1]Science and Technology Group, Okinawa Institute of Science and Technology Graduate University (OIST), 1919-1 Tancha, Onna-son, Okinawa 904-0495, Japan. [2]Scientific Imaging Section, OIST, 1919-1 Tancha, Onna-son, Okinawa 904-0495, Japan. [3]Instrumental Analysis Section, OIST, 1919-1 Tancha, Onna-son, Okinawa 904-0495, Japan. [4]These authors contributed equally: Hinako Tamotsu, Koji Koizumi. ✉e-mail: reina.komiya@oist.jp

somatic cell layer and germ cell. Based on these results, we revealed the site-specific regulation of three AGOs, AGO1b/d, and MEL1, in the anther phasiRNA pathway in rice.

## Results

### Seed sterility with abnormal anther development via *AGO1b* and *AGO1d* mutation

To elucidate the reproductive roles of AGO1b and AGO1d, we performed genome editing of *AGO1b* (Os04g0566500) or *AGO1d*

(Os06g0729300) with the CRISPR/Cas9 system, using unique sequences of the 1st exon as a guide RNA. We obtained three different allele mutants for each of *AGO1b* and *AGO1d*, namely *ago1b-1*, *ago1b-2*, *ago1b-3*, *ago1d-1*, *ago1d-2*, and *ago1d-3* (Fig. 1a and Supplementary Fig. 1). Each of the six mutants showed slightly reduced seed fertility compared to that in the *japonica* rice 'Nipponbare' control, with no obvious effects in the mature anthers or pollen of any individual mutant (Supplementary Fig. 2). *ago1b-1* and *ago1d-3* were used for the following analysis. Using Wes as an automated western analysis

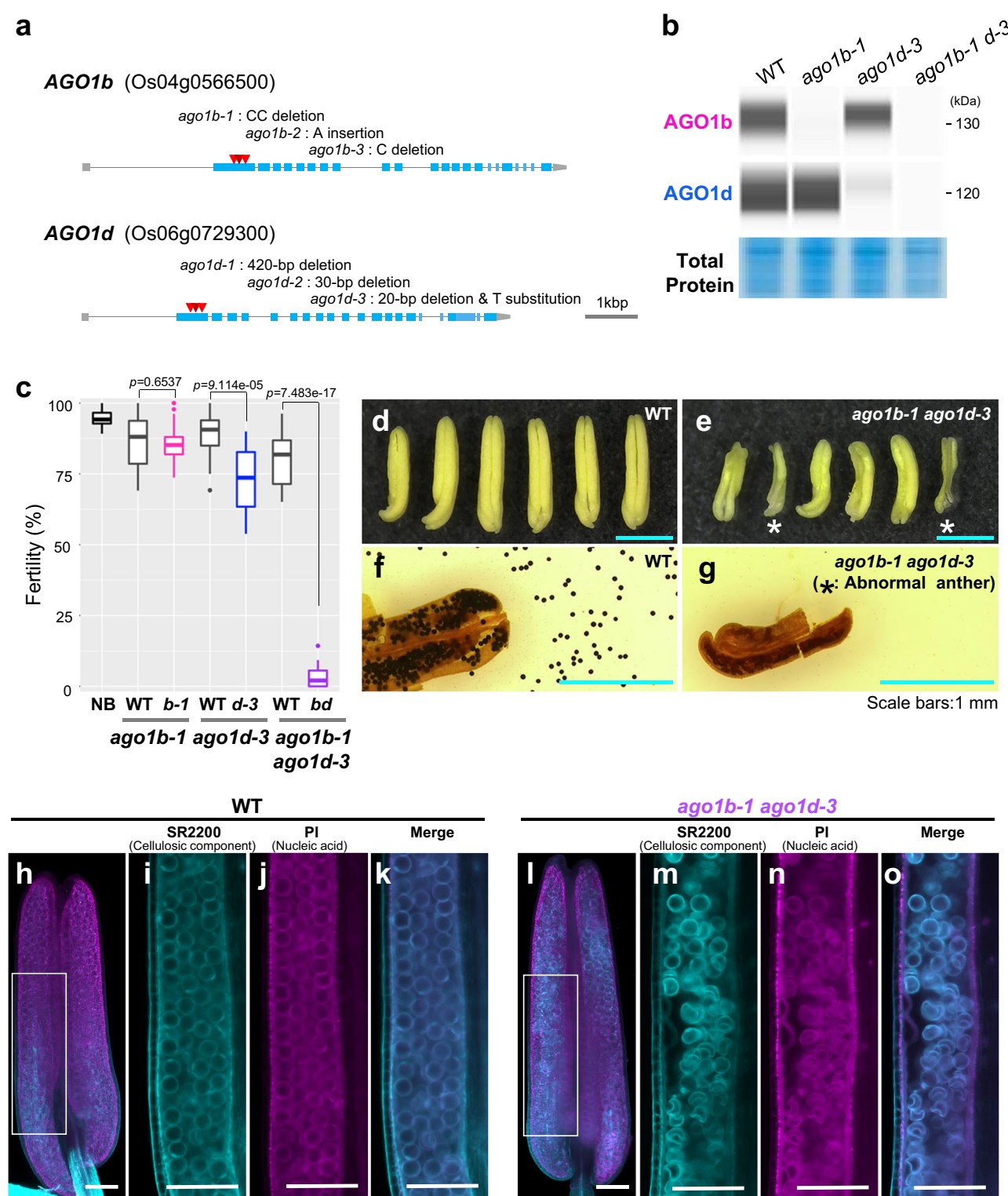

**Fig. 1 | Seed sterility with abnormal anther development via AGO1b and AGO1d mutation. a** Schematic structures of *AGO1b* and *AGO1d* genes, and three alleles for each of the *ago1b* and *ago1d* genome-editing mutants. Red triangles represent the deletion or insertion loci of *ago1b-1*, *ago1b-2*, *ago1b-3*, *ago1d-1*, *ago1d-2*, and *ago1d-3*. **b** Wes analysis of AGO1b and AGO1d in Nipponbare (WT), *ago1b-1*, *ago1d-3*, and *ago1b-1 ago1d-3* double mutants. Total proteins were extracted from 0.5-mm anthers and stained with CBB in the bottom panel. The experiment was performed twice with similar results. **c** Fertility of Nipponbare (NB), WT segregated from *ago1b-1* back-crossed once with NB, *ago1b-1*, WT segregated from *ago1d-3* backcrossed once with NB, *ago1d-3*, WT segregated from *ago1d-3* backcrossed once with *ago1b-1*, and *ago1b-1 ago1d-3* mutants. In each box plot, the center line indicates the median and the edges of the box represent the first and third quartiles. Whiskers indicate 1.5× the interquartile range. Points are plotted as outliers. *n* = 16 panicles from 4 plants (NB),

21 panicles from 5 plants (WT for *b-1*), 23 panicles from 6 plants (*b-1*), 19 panicles from 5 plants (WT for *d-3*), 16 panicles from 5 plants (*d-3*), 13 panicles from 4 plants (WT for *b-1 d-3*), and 11 panicles from 5 plants (*b-1 d-3*). Source data are provided with this paper. *P* values were calculated by the two-sided Student's *t* test. **d, e** Mature anthers of WT and *ago1b-1 ago1d-3* double mutants. asterisk indicates severely abnormal anthers with varied sizes and shapes. **f, g** Mature pollen grains of WT and *ago1b-1 ago1d-3* double mutants stained with iodine-potassium iodide. The pollen staining was performed four times for WT and seven times for the *ago1b-1 ago1d-3* double mutants with similar results. **h–o**. 3D imaging of anthers of WT and *ago1b-1 ago1d-3* double mutants, which were stained with SCRI Renaissance 2200 (**i**, **m**) and PI (**j**, **n**). **i–k**, **m–o** Slice images of the anther before 3D reconstruction of the **h** and **l** images. Laser excitation/emission: 405/BP_420–470 nm (SR2200), 561/LP_585 nm (PI). Images are shown as merged images of SR2200 and PI staining (**h**, **k**, **l**, **o**).

system, we found a specific reduction of AGO1b protein in *ago1b-1* mutants, which have a 2-bp CC deletion in the 1st exon of the *AGO1b* genomic region. In the *ago1d-3* mutants, with both a 20-bp deletion and a T substitution, AGO1d protein was specifically decreased, but not AGO1b (Fig. 1b). We next also generated double mutants of AGO1b/d in which *ago1b-1* pollen was used to pollinate the *ago1d-3* plant. Both AGO1b and AGO1d proteins were reduced in *ago1b-1 ago1d-3* double mutants (Fig. 1b). *ago1b-1* single mutant backcrossed once with Nipponbare showed almost normal fertility when compared to the fertility of segregating wild type (WT). In contrast, *ago1d-3* backcrossed once with Nipponbare showed partial sterility. Interestingly, *ago1b-1 ago1d-3* double mutants exhibited severe sterility compared to segregating WT (Fig. 1c). Furthermore, the anthers of *ago1b-1 ago1d-3* double mutants were more curled and shorter, and with more varied sizes and shapes, than WT anthers (Fig. 1d, e). Mature pollen retained in the abnormal anthers lacked starch in severe-type anthers of this double mutant, while some pollen retained in the semi-abnormal anthers was stained (Fig. 1e, g and Supplementary Fig. 3).

To differentiate the internal structure of the mutant anthers, we further observed them after double staining with SR2200 and propidium iodide (PI) using 3D imaging methods[28]. Even the *ago1b/d* double mutant, which displayed normal anther structure, showed abnormalities in the internal structures of the anther: the pollen grains were not filled in parts of the anther locule, some grains were crushed, and the inner layer development was partially abnormal in 0.9-mm anthers (Fig. 1h–o). These mutant analyses demonstrate that AGO1b/d redundantly regulate anther development, thus affecting the alignment, filling, and development of pollen during anther wall development, which are different from the meiotic progression function of germ cell-specific MEL1.

Next, we performed RNA sequencing using total RNAs extracted from 0.5-mm anthers of WT and the *ago1b-1 ago1d-3* mutant (Supplementary Data 1). A total of 1,894 genes were upregulated in the *ago1b-1 ago1d-3* double mutant, while 784 were downregulated, relative to their expression in WT plants (Supplementary Fig. 4a, b; false discovery rate (FDR) < 0.05, fold change (FC) > 5). Two upregulated genes in *ago1b/d* double mutants, LOC_Os08g29669 and LOC_Os02g02870, are among nine phasiRNA target genes, whose mRNAs have been demonstrated to be cleaved by 21-nt phasiRNAs using meiocyte-based PARE/degradome sequencing[13]. There were also 111 upregulated *21PHAS*s, which are precursors of 21-nt phasiRNAs, and 35 down-regulated *21PHAS*s in the *ago1b/d* double mutant (Supplementary Fig. 4c, d; FDR < 0.05, FC > 2). The high numbers of upregulated genes and *21PHAS*s suggest that AGO1b and AGO1d function in silencing.

**Interaction of 21-nt U-phasiRNAs with AGO1b and AGO1d**

Wes using anthers revealed that both AGO1b and AGO1d were enriched from the early meiosis stage to the post-meiosis stage among six stages of 0.4–0.9-mm anther development (Fig. 2a); each stage is explained in relation to germ and soma development in Supplementary Data 2. Next, to identify the small RNAs interacting with AGO1b and AGO1d, we performed RIP using 0.4–0.9-mm anthers from pre-meiosis to

post-meiosis. AGO1b and AGO1d were identified using RIP fractions in Wes analysis (Fig. 2b). Mass spectrometry analysis confirmed that AGO1b was immunopurified in AGO1b-RIP fractions, and AGO1d in AGO1d-RIP fractions, using antibodies generated in rabbits and mice, respectively (Supplementary Data 3). Next, the RNAs extracted from each of the AGO1b/d complexes were sequenced (Supplementary Data 4). AGO1b and AGO1d bound mainly to 21-nt small RNAs that have 5′-terminal uracil (U-phasiRNAs) (Fig. 2c, d). Additionally, the majority of AGO1b/d–small RNAs are categorized as 21-nt phasiRNAs derived from *21PHAS*s, which are anther-specific lncRNAs (Fig. 2e). miRNAs interacting with AGO1b/d occurred at low frequency: 4.2% for AGO1b and 5.6% for AGO1d. In the AGO1b/d–miRNA group, AGO1d tended to bind to the 18 family members of miR2118, from miR2118a to miR2118r, more so than AGO1b (Fig. 2f). Argonaute proteins are known to sort small RNAs having specific 1st nucleotides in plants[26,29]. The results of analysis of 21-nt U-phasiRNA loading onto anther AGO1b/d coincide with the size and 5′-terminal nucleotide frequency of small RNAs interacting with AtAGO1; however, reproductive AGO1b/d–phasiRNAs are clearly distinct from the AtAGO1–miRNAs and vegetative OsAGO1 subfamily–miRNAs (Fig. 2)[25,26].

**Segregation of U-phasiRNA subpopulations between AGO1b and AGO1d**

Next, we performed clustering analysis using the *PHASIS* program[30] to detect the genomic loci of *21PHAS*s, which are the origin of 21-nt phasiRNAs loaded onto AGO1b/d. Thus, we identified 2337 clusters of AGO1b-phasiRNAs and 2441 clusters of AGO1d-phasiRNAs in the rice genome (Fig. 3a and Supplementary Data 5 and 6). In contrast to the 21-nt phasiRNA clusters, we found 44 clusters in AGO1b–24-nt phasiRNAs and 73 clusters in AGO1d–24-nt phasiRNA (Fig. 3a and Supplementary Data 7 and 8). In the 21-nt phasiRNA clusters, the majority of AGO1b–phasiRNA clusters overlapped the loci of AGO1d–phasiRNA clusters (Fig. 3a, b).

Our previous study identified 1,345 loci for *21PHAS*s, termed 21-nt phasiRNA clusters, using total small RNAs extracted from 0.5-mm anthers[27]. We performed heatmap analysis of small RNAs interacting with AGO1b or AGO1d based on these 1345 *21PHAS* loci and could classify the phasiRNAs into mainly two types of clusters, AGO1b-type and AGO1d-type clusters (Fig. 3c). AGO1b interacts predominantly with 21-nt phasiRNAs in the AGO1b-type clusters, while AGO1d interacts pre-dominantly with those in the AGO1d-type clusters (Fig. 3c–e), suggesting that AGO1b/d also segregate subpopulations of the U-phasiRNAs.

**Nuclear AGO1b and cytoplasmic AGO1d in soma and germ**

The anther is a major part of the male reproductive organ in plants, consisting of somatic anther walls and germ cells, known as the pollen. The somatic anther walls consist of four cell layers (epidermis, endo-thecium, middle layer, and tapetum) during early meiosis, in which stages AGO1b and AGO1d are abundant (Fig. 2a and Supplementary Data 2). Recently, we developed a 3D-anther immunostaining system at subcellular and single-cell resolution[31]. To investigate the localization

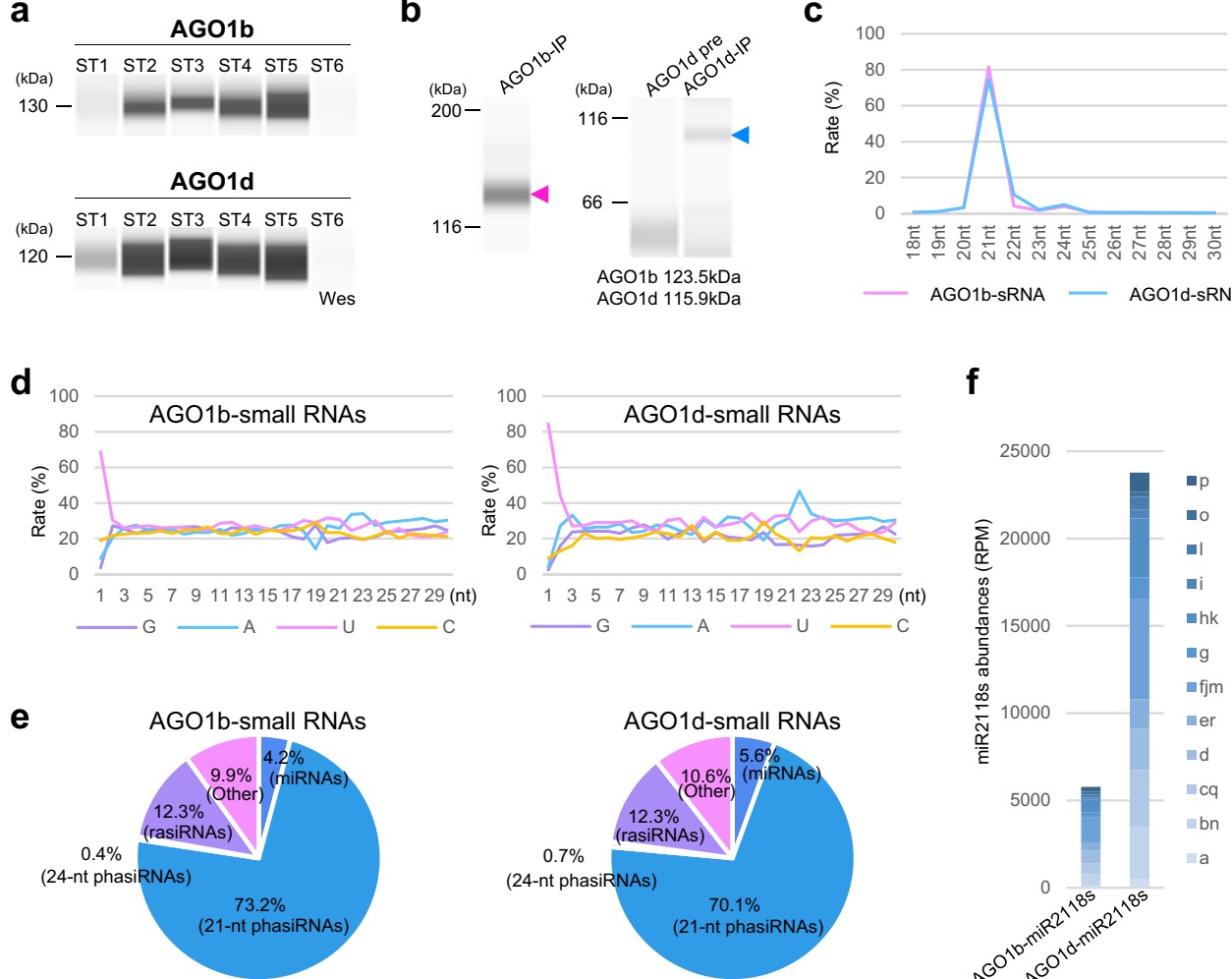

**Fig. 2 | Association of 21-nt U-phasiRNAs with AGO1b and AGO1d. a** Wes analysis of AGO1b and AGO1d proteins in developing anthers. AGO1b and AGO1d total proteins, extracted from stage 1 (ST1), stage 2 (ST2), stage 3 (ST3), stage 4 (ST4), stage 5 (ST5), and stage 6 (ST6) anthers, were enriched during meiosis and post-meiosis (stages 2–5). ST1 is pre-meiosis, or primordial germ cell initiation; ST2–4 are during meiosis; ST5 is the microspore development stage; and ST6 is the bicellular pollen stage. The wes using ST1, ST2, and ST3 total proteins was repeated with similar results. **b** Wes images of AGO1b/d RIP fractions with anti-AGO1b or -AGO1d. Arrowheads show the size distribution of AGO1b protein (magenta) and AGO1d protein (blue). The RIP analysis was performed three times with similar results. **c** Size distribution of AGO1b–small RNAs and AGO1d–small RNAs. **d** Relative frequency of each nucleotide in the AGO1b/d–small RNAs. **e** Pie charts summarizing the source of AGO1b/d–small RNAs. AGO1b and AGO1d mainly associate with 21-nt U-phasiRNAs derived from *21PHAS*s/lncRNAs. **f** Read counts of miR2118 family members in AGO1b–small RNAs and AGO1d–small RNAs. The reads data show the average counts of two replicates for AGO1b–small RNAs and four replicates for AGO1d–small RNAs.

of these AGO1b/d proteins in anthers, we performed the 3D multiple immunostaining against AGO1b and AGO1d using whole-mount anthers at early meiosis. We captured continuous image data of the anther, consisting of 80 slices with 0.6-μm intervals from the outer epidermis to the inner pollen mother cells (PMCs) in longitudinal Z sections, and created a movie by stacking these images (Supplementary Movie 1). We thus were able to detect the localization of AGO1b/d proteins in each cell layer and in PMCs of rice anthers at this stage (Supplementary Movie 1 and Fig. 4). It is also possible to distinguish the four somatic cell layers in X sections, which are cross-sections of anthers (Supplementary Movie 1, upper figure).

Both AGO1b and AGO1d were enriched in the nucleus and at the cell membrane of the epidermis, whose cells are the largest in the anther wall layers (Fig. 4a–d). In the elongated-transverse endothecium cells, AGO1d was more abundant in cytoplasm near the nucleus (Fig. 4e–h). In the cells of the middle layer, AGO1b was present in the nucleus. However, weak fluorescence signals of AGO1d were detected in the cytoplasm in the middle layer cells (Fig. 4i–l). Furthermore, AGO1b localization in the tapetum layer was restricted to the nucleus,

while AGO1d was present in both nucleus and cytoplasm (Fig. 4m–p). In somatic anther walls, AGO1b thus tends to localize in the nucleus, and AGO1d in the cytoplasm. The differing intracellular localization of AGO1b in the nucleus and AGO1d in the cytoplasm of all four cell types in the anther wall concurs with the segregation of phasiRNA loading onto AGO1b and AGO1d (Figs. 3c–e and 4a–p; Supplementary Movie 1). Moreover, AGO1b was mainly detected in the nucleus of PMCs at early meiosis (Fig. 4r), whereas AGO1d was present in both nucleus and cytoplasm in germ cells (Fig. 4s).

**Comparative spatial and intracellular distribution of AGO1b/d and MEL1 in anthers**

To investigate in more detail the cellular and subcellular localization of three reproductive AGOs in anthers, we first generated antibodies against MEL1 in guinea pigs. Automated western analysis using Wes, with these MEL1 antibodies, confirmed that a protein with the predicted molecular mass of MEL1 was present at the early meiotic stage, when the anthers are 0.5-mm long (Supplementary Fig. 5). Furthermore, mass spectrometry analysis confirmed that MEL1 was

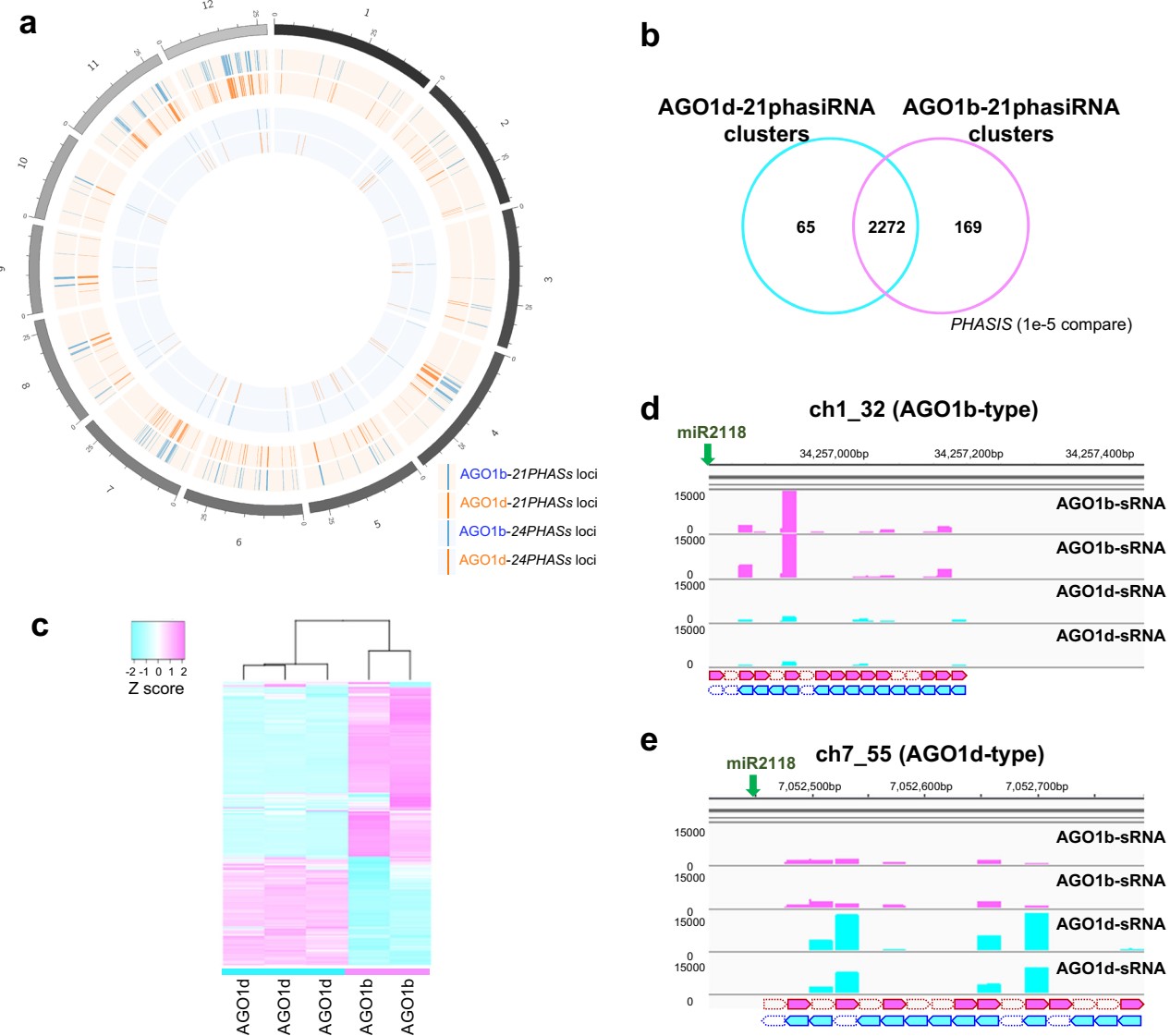

**Fig. 3 | AGO1b and AGO1d sub-grouped segregation in U-phasiRNA sorting.**
**a** Circular plot showing the distribution of the *21PHAS*s or *24PHAS*s on the 12 rice chromosomes from which 21-nt or 24-nt phasiRNAs interacting with AGO1b or AGO1d were derived. *21PHAS* loci, where 21-nt phasiRNAs associating with AGO1b (blue) and AGO1d (orange) are produced, are displayed in the outer pale orange zone. *24PHAS* loci, where 24-nt phasiRNAs associating with AGO1b (blue) and AGO1d (orange) are produced, are displayed in the inner pale blue zone. **b** Venn diagram of overlapping 21-nt phasiRNA clusters, *21PHAS* genomic loci that were identified using small RNAs that sorted into AGO1b or AGO1d. **c** Heatmap of predominant pattern of small RNAs interacting with AGO1b and AGO1d. **d, e** Small RNA-seq reads of AGO1b−small RNAs and AGO1d−small RNAs (two replicates of each) that were mapped to the AGO1b-type clusters (**d**) and AGO1d-type clusters (**e**).

immunopurified in MEL1-RIP fractions, using this antibody (Supplementary Data 9). We next performed 2D four-color immunostaining against AGO1b, AGO1d, and MEL1, as well as DAPI nuclear staining with cells released from 0.5-mm anthers. Using a confocal microscope, we captured image data for the PMCs or somatic cells, whose origin from a specific somatic cell type (Ep, En, Ml, or Ta) remains unknown because the anthers were crushed during sample preparation. AGO1b and AGO1d were abundant in both somatic anther wall cells and PMCs, while MEL1 was restricted to germ cells, PMCs (Fig. 5a–h). These AGO fluorescence signals were not detected in the negative control samples, which were not treated with primary antibodies for AGO1b/d or MEL1, but only with secondary antibodies (Supplementary Fig. 6). The detection only of DAPI signals in the negative controls indicates that the fluorescence from immunostaining reflects the localization of respective AGOs, and is not autofluorescence.

Furthermore, AGO1b was enriched in the nucleus of PMCs, while MEL1 was abundant in the cytoplasm of PMCs (Fig. 5f, h)[6]. In contrast, AGO1d was present in both the nucleus and cytoplasm in germ cells

(Figs. 4s and 5g). Proper intracellular localization, in addition to cell specificity, among AGO1b/AGO1d/MEL1 may be required for germ cell development (Fig. 5a–h).

High-resolution observations showed that AGO1b and AGO1d formed foci in PMCs (Fig. 5i–k), suggesting that AGO1b/d−phasiRNAs formed RNA granules in germ. Additionally, most of the cytoplasmic foci of these AGO1b−small RNA complexes did not colocalize with the foci of the AGO1d complexes (Fig. 5l). The different positioning of foci, in addition to the difference in cellular and intracellular localization between AGO1b and AGO1d, may be crucial for their individual functions during anther development.

### U-phasiRNA-carrying AGO1b/d as mobile signals from anther wall to PMCs

Next, we performed in situ hybridization (ISH) to investigate the RNA localization of *AGO1b*, *AGO1d*, and *MEL1* in anthers from 2–2.5-mm inflorescences. The ISH data enable us to distinguish the four cell layers of the anther wall, revealing that the middle layers are thinner than the

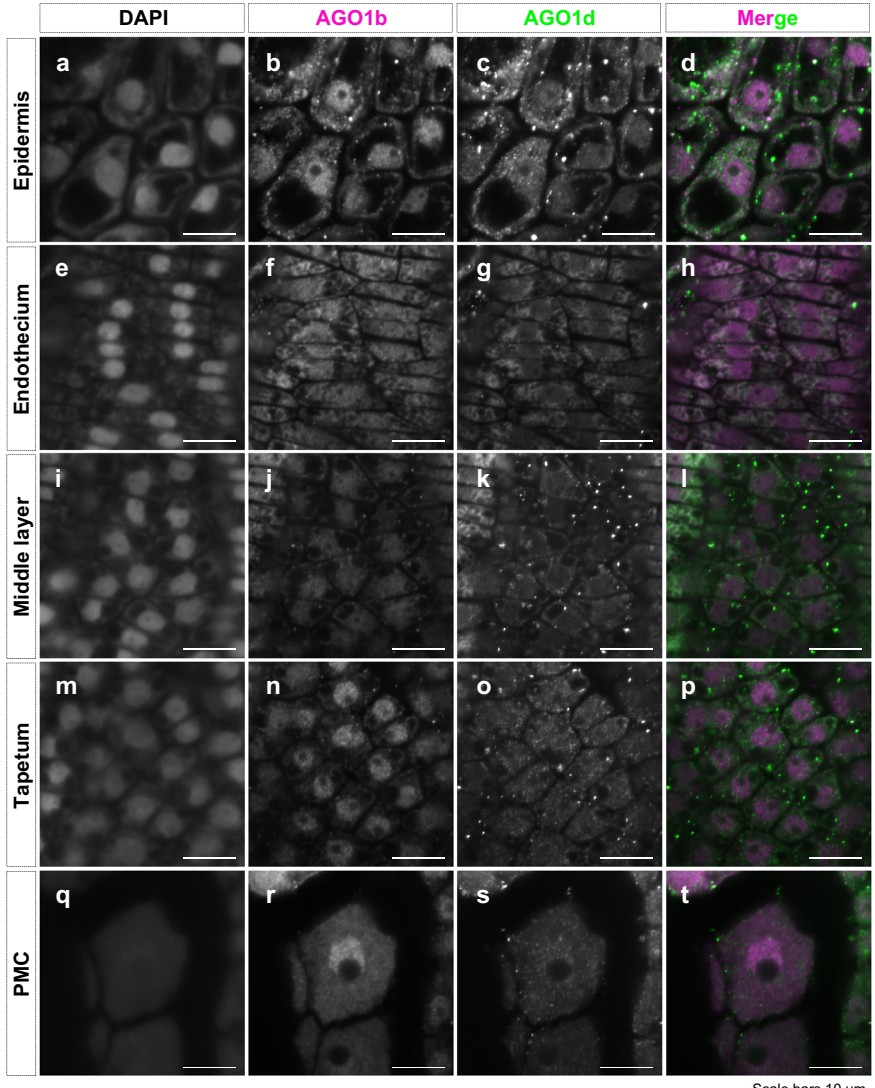

Scale bars 10 μm

**Fig. 4 | Nuclear AGO1b and cytoplasmic AGO1d in soma and germ.**
**a–t** Immunostaining using whole 0.5-mm anthers at early meiosis (stage 2) against AGO1b (magenta, bfjnr) and AGO1d (green, cgkos). DAPI staining was used as a nucleolus marker (aeimq). AGO1b and AGO1d fluorescence images were merged (dhlpt). This (3D)-immunostaining enables us to distinguish the cell types as epidermis (**a–d**), endothecium (**e–h**), middle layer (**i–l**), tapetum layer (**m–p**), and pollen mother cells (PMC) (**q–t**). The 3D immunostaining using whole anthers was performed three times with similar results. Laser excitation/emission are 405 nm/ 410–455 nm for DAPI, 488 nm/490–552 nm for AGO1d, and 561 nm/544–615 nm for AGO1b.

tapetum. Thus, the stage is between the premeiotic S-phase and the early meiosis. *AGO1b* mRNA was detected in the endothecium and middle layers of somatic anther walls (Fig. 6a and Supplementary Fig. 7a, b). *AGO1d* mRNA also occurs in the somatic anther walls, more markedly in the tapetum, at this stage (Fig. 6b and Supplementary Fig. 7c, d). *MEL1* localization is restricted to PMCs, as expected: *MEL1* mRNA and MEL1 protein are well-known PMC-specific markers (Fig. 6c and Supplementary Fig. 7e)[5,6]. *AGO1b* and *AGO1d* mRNAs were rarely detected in PMCs, while *MEL1* was specifically expressed in germ cells (Fig. 6a–c).

Moreover, qPCR using RNAs extracted from PMCs showed that *MEL1* was highly expressed, in contrast to extremely low expression of *AGO1b* and *AGO1d* (Fig. 6d). These results indicate that there is tissue-specific transcriptional regulation between somatic *AGO1b/d* and germ *MEL1* in anthers.

The 2D/3D immunoimaging results also indicate that AGO1b is enriched in PMCs in addition to being localized in the tapetum cell layer (Figs. 4n, r and 5b, f). However, ISH showed that *AGO1b* mRNA was expressed in the endothecium and middle layers of the anther wall, but rarely in the inner tapetum layer and PMCs (Fig. 6a). Likewise, AGO1d was localized in the PMCs and anther walls, while *AGO1d* mRNA

was expressed specifically in the anther walls (Figs. 4–6). The different localization of mRNA and protein of *AGO1b/d* subfamily genes in the anther suggests that AGO1b and AGO1d migrate from outer layers to inner layers and PMCs as carriers of phasiRNA.

## Discussion
In summary, we have uncovered novel RNA-induced silencing complexes (RISCs) comprising AGO1b/d–U-phasiRNA, which are enriched in both soma and germ in anthers, and specifically localized to the nucleus in PMCs; these RISCs differ from the germ-specific RISC with cytoplasmic localization by MEL1–C-phasiRNAs (Fig. 6e). Of particular interest is the discrimination of subcellular localization in anther wall layers, where AGO1b is enriched in the nucleus and AGO1d is mainly localized in the cytoplasm. In addition to this segregation of AGO1b/d localization in soma, the differences between MEL1, AGO1b, and AGO1d in germ subcellular localization may be critical for the function of C-phasiRNAs and U-phasiRNAs. Recent studies, including the trans-acting silencing of MEL1–C-phasiRNAs via cleavage in germ[13,14], reflect the possibility that AGO1b/d–U-phasiRNAs in the nucleus act as decoys, similar to the function of *Arabidopsis* AGO10[32]. Alternatively, AGO1b/

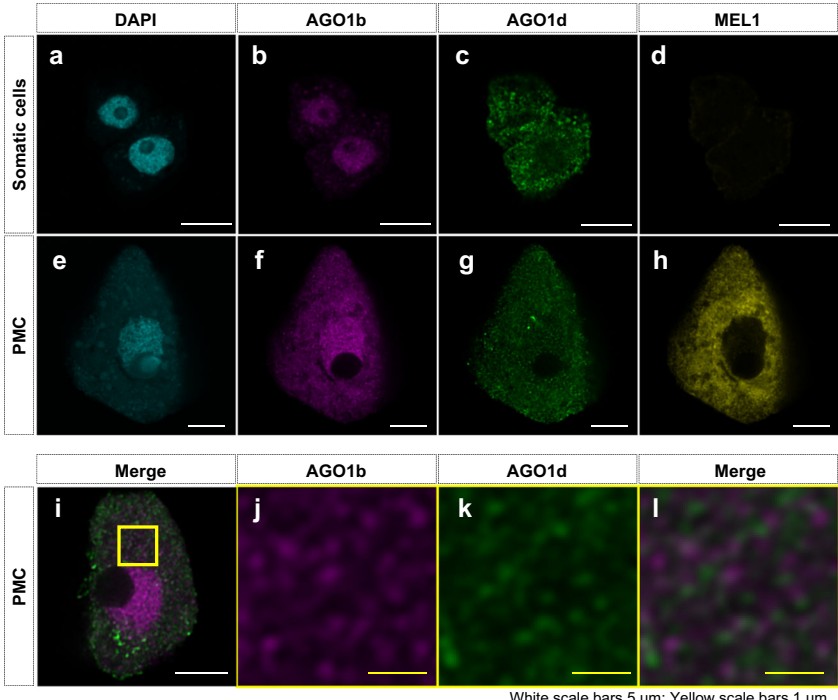

White scale bars 5 μm; Yellow scale bars 1 μm.

**Fig. 5 | Comparative spatial and intracellular distribution of AGO1b/d and MEL1 in anthers. a–h** Immunostaining using anti-AGO1b, anti-AGO1d, and anti-MEL1, as captured with the Airyscan function of LSM 880. AGO1b/d are abundant in both somatic anther wall cells and PMCs from the 0.5-mm anthers (magenta/green) (**b, c, f, g**). However, MEL1 is restricted to PMCs (**d, h**). Additionally, there is a difference in intracellular localization of nuclear AGO1b, cytosolic MEL1, and nuclear/cytosolic AGO1d in PMCs (**e–h**). Cyan signals indicate DAPI (**a, e**). **i–l** Immunoimaging with anti-AGO1b and anti-AGO1d in PMCs. Enlarged images of the cytoplasm of PMCs (**j–l**). The primary AGO1b foci (magenta) do not overlap with the AGO1d foci (green) of germ cytoplasm (**l**). The immunostaining was performed three times with similar results.

d–U-phasiRNAs may have a novel cis-epigenetic function during meiosis, as is the case for cis-regulation of 24-nt meiosis phasiRNAs with CHH DNA methylation[16]. Our results on cell type- and intracellular-specific AGO localization in the rice male organ point to the importance of AGO combination-mediated RNA silencing during reproduction. The subcellular- or speckle-specific degradome and/or methylome will be required for understanding the silencing roles of RISCs in controlling anther development. Epigenetic regulation of anther-specific AGO2 precisely initiates programmed cell death in tapetum development, resulting in normal pollen development in rice[33]. Furthermore, phosphorylation or ubiquitination of reproductive AGOs is vital for temperature-dependent male fertility or meiosis to ensure accurate reproduction via phasiRNA control in crop species[34,35]. Therefore, understanding reproductive AGOs' functions and combinations should lead to the elucidation of the molecular roles of phasiRNAs in male organ development, which has critical implications for stable crop yield.

We found that the AGO1b/d proteins were abundant in tapetum and PMCs, while *AGO1b* mRNA level was extremely low in these cell types and *AGO1d* mRNA was also rarely detected in the PMCs (Figs. 4–6). *ago1b-1 ago1d-3* double mutants additionally exhibited defects of alignment, filling, and development of pollen even though the 3D structure of anthers appeared normal (Fig. 1h–o). Thus, AGO1b/d–U-phasiRNA RISCs may have a non-cell-autonomous function via soma-to-germ redistribution during anther development. In contrast, MEL1–C-phasiRNA RISC causes trans-acting silencing in meiotic progression. The combined function of meiotic MEL1–C-phasiRNA RISC and soma-to-germ AGO1b/d–U-phasiRNA RISC is thus required for proper anther development in rice. Recent reproductive studies, revealing the cell-to-cell transportation of 24-nt phasiRNAs and the regulation of pollen development by somatic transcription factors, also imply non-cell-autonomous regulation in anther development[18,36]. Differential localization between *AGO1d* mRNA and AGO1d protein,

which functions in temperature-dependent anther development, has been reported[37,38]. We therefore propose that AGO1b/d constitute a mobile signal from the anther wall to the PMCs as carriers of U-phasiRNAs (Fig. 6e). Mobile protein signals including florigen and homeobox transcription factors are essential for accurate development and cell fate determination in higher plants[22,39–41]. Understanding the nature of reproductive AGO–phasiRNA silencing via cell-to-cell communication should provide new insights into the non-cell-autonomous mechanism in anther development and could greatly enhance the reproductive competence of plants.

## Methods

### Plant materials, growth conditions, and fertility rate

Rice (*Oryza sativa* L., subspecies *japonica*, cultivar Nipponbare) was used in this study. Plants were grown in growth chambers at 70% humidity under long day (LD) conditions, with a daily cycle of 14 h of light at 29.5–30 °C and 10 h of dark at 25 °C, for the mutant analysis in Fig. 1. To align the sampling stages, plants were grown for 40 days under LD conditions and then transferred to short day conditions (10 h of light and 14 h of dark) until harvest for the data in Figs. 2–6.

The numbers of all seeds and fertile seeds were counted in each panicle, and we calculated the fertility rate per panicle as fertile seed numbers/total seed numbers. One–nine panicles were counted for one biological replicate plant. Finally, fertility data shown in box plots are from more than three biological replicates for each mutant/control line (Nipponbare, WT, *ago1b-1*, *ago1b-2*, *ago1b-3*, *ago1d-1*, *ago1d-2*, *ago1d-3*, and *ago1b-1 ago1d-3*) for Fig. 1c and Supplementary Fig. 2a. The fertility rate of each panicle is shown in Source Data.

### Generation of AGO1b, AGO1d, and MEL1 antibodies

A synthetic peptide, AGO1b (Cys-GSSQRAERGPQQH-OH), was used to raise rabbit polyclonal antibody against rice AGO1b.

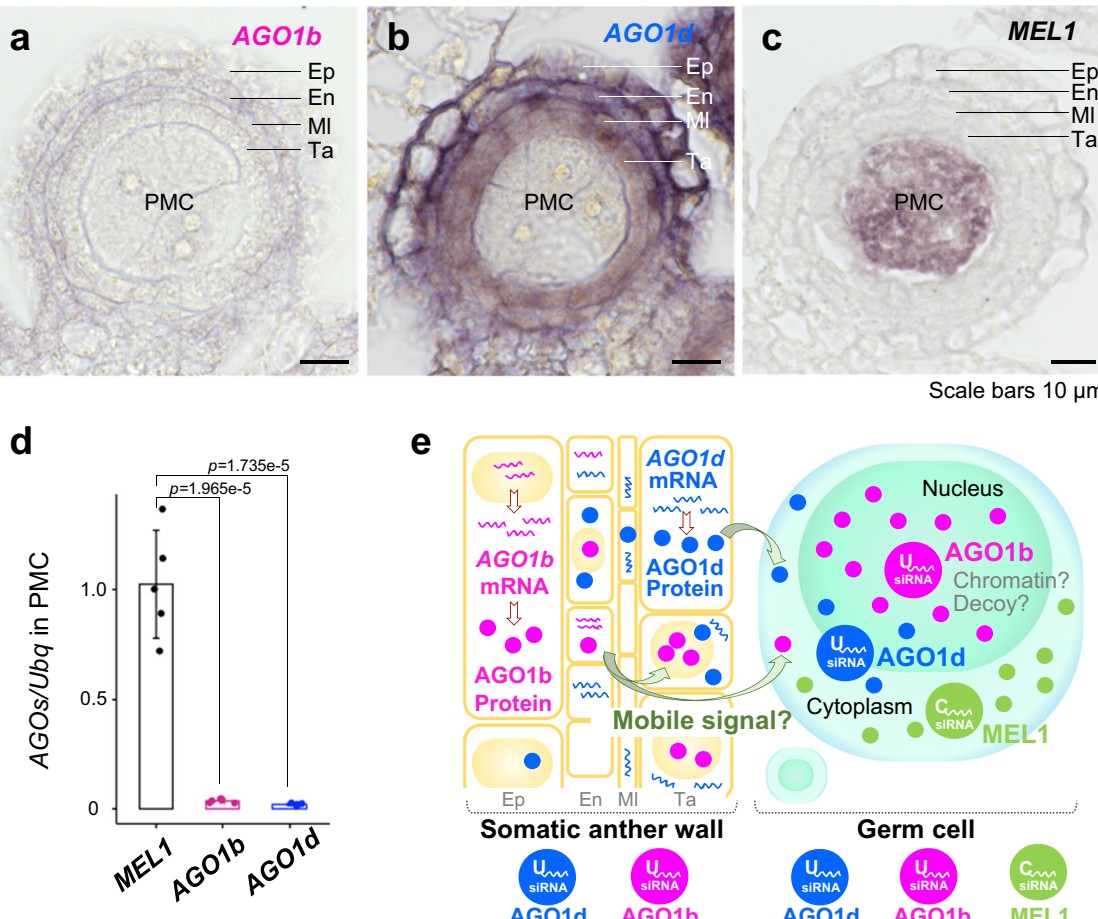

**Fig. 6 | U-phasiRNA-carrying AGO1b/d as mobile signals from anther wall to PMCs. a–c** In situ hybridization of *AGO1b*, *AGO1d*, and *MEL1* probes using anthers from 2.0–2.5-mm inflorescences. *AGO1b* localization was more highly enriched in endothecium (En) and middle layer (Ml) than in tapetum layers (Ta) between pre-meiotic S-phase and the early meiosis. **a** *AGO1d* expression is abundant in anther wall, especially tapetum (Ta), middle layer (Ml), and endothecium (En) (**b**), while MEL1 is restricted to the PMCs (**c**). These images (**a–c**) are enlargements from one anther locule of images in Supplementary Fig. 7 (**a, c, e**). The in situ hybridization was performed three times for *AGO1b* and twice for *AGO1d* with similar results. Four anthers were imaged for each in situ hybridization (*n* = 12 anthers (*AGO1b*), 8 anthers (*AGO1d*), 4 anthers (*MEL1*)). **d** qPCR of *AGO1b*, *AGO1d*, and *MEL1* using total

RNAs extracted from PMCs. *MEL1* is highly expressed in PMCs, while expression of both *AGO1b* and *AGO1d* is extremely low. The bar represents mean ± SD (*n* = 5). Student's *t* test (two-sided). Source data are provided with this paper. **e** Model of rice site-specific RNA silencing in rice anther development. *AGO1b* and *AGO1d* mRNA are increased in the somatic anther wall during pre-meiosis and early meiosis and translated; AGO1b/d proteins interacting with 21-nt U-phasiRNAs then move and become enriched in the tapetum and PMCs, acting as a mobile signal. AGO1b/d–U-phasiRNAs are detected in both soma and germ, and thus differ from MEL1–C-phasiRNAs in germ. Moreover, their intracellular localization differs in PMCs: cytoplasmic for MEL1, nucleolar for AGO1b, and both cytoplasmic and nuclear for AGO1d.

---

Oligopeptides AGO1d-1 (Cys-GRGSYYPQAQQYH-OH) and AGO1d-2 (Cys-HQQPYNSSVRPQH-OH) were used to raise antibodies in rabbits and mice. Two oligopeptides, MEL1-1 (Cys-CVYGAPMPAAHHQGAYQ-OH) and MEL1-2 (Cys-GQAVAREGPVEVRQLPKC-OH), were used to raise antibodies in guinea pigs. Animals were immunized five times with keyhole limpet hemocyanin–peptide conjugates mixed with Freund's complete adjuvant, and were bled seven days after the last immunization. Antiserum was purified on a peptide coupling purification column. SCRUM Inc. performed all procedures for antibody generation. The animal experimentation committee of SCRUM Inc. follows its own established guidelines for animal care.

### Western analysis
Wes is an automated western analysis using capillaries, not gel blotting. Results are shown in a digital image resembling a western blotting analysis. Total proteins were extracted from 0.4–0.9-mm anthers of the WT (Fig. 2a), and 0.5-mm anthers of the WT, *ago1b-1*, *ago1d-3*, and *ago1b-1 ago1d-3* double mutants (Fig. 1b). The anthers were ground and mixed with extraction buffer (150 mM NaCl, 50 mM Tris–HCl (pH 7.5), 0.1% (v/v) Tween 20, 10% (v/v) glycerol, 1 mM dithiothreitol (DTT), 1 mM

Pefabloc SC (Roche), 1× Complete Protease Inhibitor Cocktail (Roche)). After centrifuging twice to remove debris, total proteins were extracted. Anti-AGO1b/d (1/20 dilution) were used as primary antibodies.

### Small RNA-immunoprecipitation
Total proteins were extracted from 0.4–0.8 mm anthers of Nippon-bare with an extraction buffer (see above). After pre-clearing using protein G-Sepharose, 1–5 μg of anti-AGO1b or AGO1d antibody was added to the sample and incubated for 1 h at 4 °C to immunoprecipate AGO1b/d–small RNA complexes. Protein G-Sepharose was added to the sample and incubated for more than 16 h at 4 °C. AGO1b/AGO1d–small RNA complexes were washed four times in wash buffer (20 mM Tris–HCl (pH 7.5), 150 mM NaCl, 5 mM MgCl$_2$, 5 mM DTT, 0.1% (v/v) NP-40, 1× Complete Protease Inhibitor Cocktail)[42].

### Small RNA or RNA sequencing
NEXTflex small RNA-seq libraries (150 bp paired-end) were prepared (PerkinElmer) for RIP-small RNA sequencing. NEBNext Ultra II directional RNA library prep kits (150 bp paired-end) were prepared (New England Biolabs) for RNA sequencing. These libraries were sequenced

using Illumina Novaseq. Two or four biological replicates were prepared for sequences of small RNAs binding to AGO1b or AGO1d, respectively (Fig. 2). Three or two biological replicates were prepared for RNA sequences of WT or *ago1b-1 ago1d-3* double mutant, respectively (Supplementary Fig. 4).

## Small RNA sequencing data analysis
Sequencing reads were trimmed using TrimGalore (http://www.bioinformatics.babraham.ac.uk/projects/trim_galore) to remove adapter sequences and sequencing bias with the following parameters: stringency = 5, quality = 20, and three prime clip R1 = 60. After trimming, reads that ranged from 18 bp to 30 bp were used. To annotate the small RNAs, perfectly matched reads were mapped onto the rice genome IRGSP1.0 (https://rapdb.dna.affrc.go.jp) with Bowtie, and the datasets of miRNA from miRbase (http://www.mirbase.org/ftp.shtml) and repeat data from RAP-DB (https://rapdb.dna.affrc.go.jp/) were used.

To identify 21- and 24-nt phasiRNA clusters, 18- to 30-nt trimmed small RNAs were analyzed using *PHASIS*[30] with default parameters. Significant clusters ($p$-value ≤ 1e$^{-5}$) from the replicates were merged. Overlapping clusters among the samples were extracted using the phasmerge function in *PHASIS*.

## RNA sequencing data analysis
Sequencing reads were mapped onto the rice genome IRGSP1.0 (https://rapdb.dna.affrc.go.jp) with HAISATII, and we counted reads with feature counts (R studio). The dataset of 1345 loci for *21PHAS*s was used[27]. iDEP.96 was used for integrated differential expression analysis (http://bioinformatics.sdstate.edu/idep96/).

## In situ hybridization
Inflorescences were fixed in FAA solution (30% ethanol, 1.85% formaldehyde, and 5% acetic acid) overnight at 4 °C and then dehydrated in a graded ethanol series. Ethanol was replaced with *t*-butyl alcohol and the samples were embedded in Paraplast Plus (Merck)[28]. The probes of the gene regions were synthesized using the Digoxygenin Labeling Kit (Roche, USA) following the manufacturer's instructions. We performed ISH for *AGO1d* localization with a previously used probe[43]. The primer sets for the probes are listed in Supplementary Data 10. Cross-sections of the tissues with 8 μm thickness were prepared using a rotary microtome. A part of the sections was stained with 0.1% (w/w) toluidine blue to identify the anthers. Sections mounted on slides were dewaxed with Lemosol (Fujifilm-Wako). Slides were incubated with 0.5 μg/mL proteinase K (Sigma) for 15 min at 37 °C, and then acetylated with 1.5% (v/v) triethanolamine in a solution of 0.25% (v/v) concentrated hydrochloric acid and 0.25% (v/v) acetic hydride. Probes (0.8 μg/mL) were used for hybridization overnight at 55 °C. The slides were washed twice with 50% (v/v) formamide in 2× saline-sodium citrate (SSC) for 30 min at 55 °C and incubated with the buffer containing 10 μg/mL RNaseA for 30 min at 37 °C to remove the unhybridized RNA probes. Next, the slides were washed twice with 2 × SSC and 0.2 × SSC for 30 min at 55 °C[28]. The blocking steps and detection of hybridized transcripts, which required anti-digoxigenin antisera conjugated to alkaline phosphatase (Roche Anti-Digoxigenin-AP, NBT/BCIP), were performed following the manufacturer's protocol. The signal was detected after 3 to 16 h of incubation at 30 °C.

## Immunostaining using whole mounts of anthers
Anthers fixed in 4% paraformaldehyde were transferred to PME buffer (50 mM PIPES, 5 mM EGTA, and 5 mM MgSO$_4$, pH 6.9) on a MAS-coated microscope slide, cut in distilled water using a scalpel, and incubated for 30 min at 25 °C. The cleaved anthers were then blocked with 3% bovine serum albumin (BSA) in PME for 60 min. The samples were placed in primary antibody solutions (rabbit anti-AGO1b or mouse anti-AGO1d, diluted 1/500 with 3% BSA in PME), degassed five times at 0.05 MPa for 2 min, and incubated overnight at 4 °C. After washing three times for

5 min with PME, the slide was placed in secondary antibody solution (Alexa Fluor 568-conjugated anti-rabbit IgG (Invitrogen, A11036) or Alexa Fluor 488-conjugated anti-mouse IgG (Invitrogen, A11001), diluted 1/200 with 3% BSA in PME), and degassed as above. The slide was incubated in a dark chamber for 2 h at room temperature, and then incubated overnight at 4 °C. The samples were washed three times with PME buffer for 5 min, incubated for 15 min at 25 °C in 1 μg/mL DAPI (Sigma, MBD0015), and washed again with PME buffer as above[31]. Samples were mounted in ProLong Gold antifade reagent (Invitrogen, P10144). Images were captured using an LSM 780 (Carl Zeiss).

## Visualization of the 3D immunostaining of the entire anthers
The images were captured using an LSM 780 (Carl Zeiss). Conditions: 40x /1.40 Plan Apochromat lens (Supplementary Movie 1) and ×63/1.46 Plan Apochromat lens (Fig. 4, Supplementary Movie 1) for detection, 405, 488, and 561 nm laser lines for DAPI, Alexa Fluor 488, and Alexa Fluor 568 excitation, 410–455 nm (DAPI), 490–552 nm (Alexa Fluor 488) and 544–615 nm (Alexa Fluor 568) filter emission. The images and animation were created using ZEN (Carl Zeiss) or Imaris 9 (Bitplane AG) software (Fig. 4 and Supplementary Movie 1).

## Immunostaining using PMCs for antibody estimation
To release the meiocytes, 4% PFA-fixed anthers were squashed in distilled water using a needle on a MAS-coated microscope slide, and incubated at 25 °C for 30 min. The meiocytes were then blocked with 3% BSA in PME buffer (50 mM PIPES, 5 mM EGTA, and 5 mM MgSO$_4$; pH 6.9) for 60 min and incubated at 4 °C overnight with rabbit anti-AGO1b, mouse anti-AGO1d, and guinea pig anti-MEL1 antibody, diluted 1/500 with 3% BSA in PME. After washing three times with PME for 5 min, the slide was incubated in a dark chamber for 3 h at 25 °C with Goat anti-Rabbit IgG (H + L) Highly Cross-Adsorbed Secondary Antibody, Alexa Fluor 568 (Invitrogen, A11036), Goat anti-Mouse IgG (H + L) Highly Cross-Adsorbed Secondary Antibody, Alexa Fluor 488 (Invitrogen, A11001), and Goat anti-Guinea pig IgG HL, Alexa Fluor 647 (abcam, ab150187), diluted 1/200 with 3% BSA in PME, followed by three 5-min washes with PME. The samples were incubated for 15 min at 25 °C in 1 μg/mL DAPI diluted 1/1000 with PME (Sigma, MBD0015), followed by washing three times with PME buffer. Samples were mounted in ProLong Gold antifade reagent (Invitrogen, P10144) and images were captured using an LSM 880 microscope (Carl Zeiss)[31].

## 3D imaging using histochemically stained anthers
Anthers fixed in 4% paraformaldehyde were stained in 0.01% (v/v) Renaissance 2200 (Renaissance Chemicals) for 1 h and 10 μg/mL propidium iodide for 1 h. Stained samples were cleared for 48–72 h in ClearSeeAlpha solution[44]. Samples were embedded in 0.1% agarose gel in a glass capillary. The images were captured using a Lightsheet microscope (Carl Zeiss Z.1) with a 20x Plan Apochromat lens.

## Mass spectrometry
The protein samples, which were AGO1b/AGO1d-IP fractions, were reduced with DTT and then alkylated with iodoacetamide and digested overnight using Lys-C/Trypsin (1:50, enzyme to protein; Promega). After terminating the digestion with 1% trifluoroacetic acid, the peptide mixture was cleaned with desalting C18 tips (StageTip, Thermo Fisher Scientific), and subsequently vacuum-dried and dissolved in 0.1% formic acid, 0.5% acetic acid in water for LC/MS analysis.

Data were collected using an Orbitrap-Fusion Lumos mass spectrometer (Thermo Fisher Scientific) coupled with the Waters nanoACQUITY liquid chromatography system. A trap column (nanoACQUITY UPLC 2G-V/M Trap 5 μm Symmetry C18, 180 μm × 20 mm, Waters) and an analytical column (nanoACQUITY UPLC HSS T3 1.8 μm, 75 μm × 150 mm, Waters) were used for chromatographic separation of samples. Peptides were separated at a flow rate of 500 nL/min using a gradient of 1–32% acetonitrile (0.1% formic acid) over 60 min. The

CHOPIN method[45] was used along with the Orbitrap-Fusion mass spectrometer using Xcalibur (v.3.0; Thermo Fisher Scientific).

Raw data files were searched against a composite target/decoy database using SEQUEST (Proteome Discoverer, v.2.2, Thermo Fisher Scientific).

Mass spectrometry analysis by Kazusa DNA Research Institute confirmed that MEL1 was immunopurified in MEL1(#1)-RIP fractions, using antibodies generated in guinea pigs (Supplementary Data 9).

## Gene targeting construction and transformation

pZH_gYSA_MMCas9 and pU6gRNA-oligo were used as vectors[46]. For editing of the *AGO1b* locus, 5′-AGCCATACTATGGCGGACCT-AGG (PAM)−3′ was used for the target sequence of the CRISPR/Cas9 vector as a single guide RNA.

5′-CTCCGGAGGCATCATCACCA-CGG (PAM)−3′ was used for genome editing of the *AGO1d* locus, and *ago1d-1* and *ago1d-2* were generated; 5′-GACCGTGGTGATGATGCCTC-CGG (PAM)−3′ was used for genome editing of the *AGO1d* locus, and *ago1d-3* was generated. Underlines show the PAM sequences. First, the oligos for guide RNAs were cloned into a pU6gRNA-oligo vector. Second, OsU6 promoter::gRNA was cut with AscI and PacI, and cloned into the pZH_gYSA_MMCas9 binary vector with hygromycin phosphotransferase selection[46]. Transgenic rice plants were generated by *Agrobacterium*-mediated transformation of rice calli (cv. Nipponbare)[47].

## RNA extraction from PMCs, cDNA synthesis, and qPCR

Six anthers with lengths of 0.5–0.7 mm were put in 10 μL direct-RT buffer (100 mM DTT and 90 mM Tris–HCl (pH 7.6))[48] on a slide (Matsunami Micro Slide Glass) and split into two parts using a scalpel, and PMCs released into the buffer were collected in a 1.5-mL tube on ice. We repeated the PMC collection procedure three more times. Next, to crush the PMCs and release RNA, we subjected the collection tube to five freeze-thaw cycles using liquid nitrogen. The buffer with crushed PMCs was used for direct cDNA synthesis with SuperScript IV reverse transcriptase and oligo dT (Invitrogen). Primers for qPCR are listed in Supplementary Data 10.

## Reporting summary

Further information on research design is available in the Nature Portfolio Reporting Summary linked to this article.

## Data availability

*AGO1b*: Os04g0566500; *AGO1d*: Os06g0729300; *MEL1*: Os03g 0800200. The gene datasets used during the current study are available in the RAP-DB repository (https://rapdb.dna.affrc.go.jp/). RIP data and RNA sequences have been deposited in the DNA Data Bank of Japan (DDBJ), under the accession codes, PRJDB13635 [https://ddbj.nig.ac.jp/resource/sra-submission/DRA015390] and PRJDB14859 [https://ddbj.nig.ac.jp/resource/sra-submission/DRA015421]. Imaging data have been deposited in the BioStudies, under the accession code, S-BSST1083 [https://www.ebi.ac.uk/biostudies/studies/S-BSST1083?key=8d7214cd-b5eb-4985-8e73-1eab99a5ccbc]. Source data are provided with this paper.

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

## Acknowledgements

This work was supported by the JST FORESTO Program (Grant Number JPMJFR204U, Japan), the JST PRESTO Program (Grant Number JPMJPR17Q3, Japan), the JSPS Grant-in-Aid for Transformative Research Areas (A) (Grant Number JP23H04752, Japan), the Naito Foundation, and the Okinawa Institute of Science and Technology Graduate University, Japan. We thank the Sequencing Section for their advice regarding sequencing, the Scientific Imaging Section for their training for the LSM 880, Dr. Jan Moren for his support of informatic analysis, Dr. Atsushi J. Nagano for his advice regarding RNA extraction from PMCs, and Ms. Saori Araki for rice cultivation and genotyping. We thank Dr. Seijiro Ono, Dr. Ian Smith, all members of the Science and Technology Group, and FORESTO members (Siomi panel) for their helpful discussions.

## Author contributions

R.K. conceived the study, conducted most of the data analysis, and wrote the manuscript. R.K. performed RIP and bioinformatics analysis of small RNAs and RNA sequences. H.T. generated *ago1b*, *ago1d*, and *ago1b ago1d* double mutants, and performed mutant phenotyping. H.T. extracted RNAs from anthers for RNA-seq and from PMCs for qPCRs, and generated the libraries. R.K. and H.T. performed 2D/3D immunostaining. K.K. and H.T. performed ISH and 3D histochemical staining for anther imaging. H.T. performed protein experiments. A.V.B. performed proteome analysis. H.T. and K.K. assisted in creating imaging figures and discussion.

## Competing interests

The authors declare no competing interests.
