## [Peer Review File · Nature Communications]

Spatial distribution of three ARGONAUTES regulates the anther phasiRNA pathwayReviewer #1 (Remarks to the Author):

The manuscript by Tamotsu et al. describes a functional characterization of two of the four rice Argonaute1 paralogs, AGO1b and AGO1d. The single mutants of AGO1b and AGO1d show little impact on rice fertility, whereas a double mutant of AGO1b and AGO1d show strongly reduced fertility. The authors further identify the small RNAs that loaded onto AGO1b and AGO1d in anthers by RIP-seq analyses and examine the cellular and sub-cellular localization of AGO1b and AGO1d. The mutant phenotype is dramatic, and the RIP-seq data provide new insights into and expand our understanding of the roles of AGO1 paralogs in loading reproductive phasiRNAs. Overall the manuscript is nicely put together, and the quality of the experiments is high. I only have some minor comments for further improvements:

Minor comments:

1. Suppl. Figure 2: It will be helpful to specify how fertility was quantified.
2. Lines 7 to 8, page 7: It is not clear which sRNA loci were used as input for the heatmap. Are these the union of the AGO1b/d-21phasiRNA loci (which would be >2,000 loci)?
3. Lines 11 to 13, page 3: the authors should clarify in which species (rice versus maize) these were observed.
4. Line 5, page 6: it's not necessary to mention the observation about SDS-PAGE, as the data are not shown.

Reviewer #2 (Remarks to the Author):

Argonaute proteins (AGOs) are essential for the epigenetic regulation of plant development. AGO1 generally binds to miRNA to function in post-transcriptional gene silencing. The authors identified the function of two AGOs, AGO1b, and AGO1d, which interact with phased small interfering RNAs (phasiRNAs) to regulate rice anther development by acting as mobile carriers of the phasiRNAs from the somatic cell layers to the germ cells in anthers. They also performed three-dimensional (3D)- immunostaining using whole anthers, which to identify the specific subcellular localization of AGO1b/d in each somatic cell layer and germ cell. The authors emphasized a reproductive RNA silencing mechanism via the specific nuclear and cytoplasmic localization of AGO1b, AGO1d, and MEL1, in the anther phasiRNA pathway of rice pollen mother cells. However, the experiments could not support their conclusions, and more solid data are needed.

Major comments:

1. The author only performed the 3D multiple immunostaining to show the differing intracellular localization of AGO1b in the nucleus and AGO1d in the cytoplasm of all four cell types in the anther wall. However, The evidence provided in the paper is insufficient and needs to be confirmed by more methods. Such as confirmed by the immune colloidal gold technique.
2. The author showed that AGO1b mRNA was expressed in the outer layers of the anther wall, and not in the inner tapetum layers or pollen mother cells. And AGO1d mRNA was also absent in the pollen mother cells by the immunostaining and RNA in situ hybridization assays, they are not enough to prove the conclusion. The quality of Fig.4 and Fig.5a-f is not good, and the RNA in situ hybridization signals cannot be clearly distinguished. Since it cannot be proved that AGO1b and AGO1d are only expressed in somatic cells, it cannot be said that they can be transferred to the mother cells.

On the other way, from the result of Tokunori Hobo et al., 2008 and Tang et al., 2010. (Tokunori Hobo et al., 2008, Various spatiotemporal expression profiles of anther-expressed genes in rice; Tang et al., 2010, Global gene profiling of laser-captured pollen mother cells indicates molecular pathways and gene subfamilies involved in rice meiosis. Plant Physiology), AGO1b, and AGO1d mRNA was expressed in the pollen mother cells. Therefore, more evidence is needed to confirm this result. Therefore, it could not be confirmed that the two genes translated into proteins and moved to the pollen mother cells.

3. The different localization of mRNA and protein of AGO1b/d subfamily genes in anther suggests that AGO1b and AGO1d migrate from outer layers to inner layers and pollen mother cells as carriers of phasiRNA. The authors did not separate the tissues from each layer of pollen mother cells for RNA-immunoprecipitation. It is not solid enough that the experimental evidence to

support this conclusion.

4. The experimental technology applied in the full text is relatively limited, actually, it does not clarify the significance and function of these two proteins to the AGO family or for the anther development in rice.

Minor comments:

1. Figure 1d, The phenotypes of the six anthers of ago1b-1 ago1d-3 are not completely consistent, what is the reason ?

2. In fact, the author did not do any experiments to prove that ARGONAUTE-mediated RNA silencing in anther development, so the title of this manuscript is not appropriate.

Responses to reviewers' comments

First of all, we would like to thank the reviewers for their constructive and valuable comments with regard to our manuscript. We have indicated the major changes made in the revised manuscript, using yellow highlighting, and provided point-by-point responses to the reviewers' comments below.

Additional results and major changes

1. 3D anther imaging to differentiate the internal structure in mutant and wild type anthers (Fig. 1h–o).
2. Transcriptome analysis using RNAs extracted from anthers of *ago1b/d* double mutants (Supplementary Fig. 4 and Supplementary Table 1) and discussion of the silencing function of AGO1b/d proteins in the main text (page 6, line 3-14).
3. Description of seed fertility rate with biological replicates in Materials and Methods (page 13, line 21 – page 14, line 3).
4. *21PHAS* information used for heatmap analysis (page 7, line 24 – page 8, line 3).
5. Generation of guinea pig anti-MEL1 antibody and confirmation of its specificity by Wes analysis (Supplementary Fig. 5).
6. Multiple immunostaining using AGO1b, AGO1d, and MEL1 antibodies with high resolution captured by LSM 880, Airyscan (Fig. 5).
7. Negative control imaging for the immunostaining (Supplementary Fig. 6).
8. New images for *in situ* hybridization with *MEL1* mRNA localization (Fig. 6c; Supplementary Fig. 7).
9. qPCR analysis of *AGO1b*, *AGO1d*, and *MEL1* expression in pollen mother cells (Fig. 6d).
10. Recent references relating to mobile AGO1d added in the Discussion (page 13, line 3-4).
11. Title changed.

REVIEWER COMMENTS

Reviewer #1 (Remarks to the Author):

The manuscript by Tamotsu et al. describes a functional characterization of two of the four rice Argonaute1 paralogs, AGO1b and AGO1d. The single mutants of AGO1b and AGO1d show little impact on rice fertility, whereas a double mutant of AGO1b and AGO1d show strongly reduced fertility. The authors further identify the small RNAs that loaded onto AGO1b and AGO1d in anthers by RIP-seq analyses and examine the cellular and sub-cellular localization of AGO1b and AGO1d. The mutant phenotype is dramatic, and the RIP-seq data provide new insights into and expand our understanding of the roles of AGO1 paralogs in loading reproductive phasiRNAs. Overall the manuscript is nicely put together, and the quality of the experiments is high. I only have some minor comments for further improvements:

Minor comments:

1. Suppl. Figure 2: It will be helpful to specify how fertility was quantified.

As the reviewer suggested, we have now described the fertility rate methods, including explanations for replicates, in Materials and Methods (page 13, line 21 – page 14, line 3) as follows. We have also provided the fertility raw data using Fig. 1c and Supplementary Fig. 2a in the source data excel file.

“Plant materials, growth conditions and seed fertility rate.

The numbers of all seeds and fertile seeds were counted in each panicle, and we calculated the fertility rate per panicle as fertile seed numbers/total seed numbers. More than three panicles were counted for one biological replicate. Finally, fertility data shown in box plots are from more than three biological replicates for each mutant/control line (Nipponbare, WT, *ago1b-1*, *ago1b-2*, *ago1b-3*, *ago1d-1*, *ago1d-2*, *ago1d-3*, and *ago1b-1 ago1d-3*) for Fig. 1c and Supplementary Fig. 2a. The fertility rate of each panicle is shown in Source Data.

2. Lines 7 to 8, page 7: It is not clear which sRNA loci were used as input for the heatmap. Are these the union of the AGO1b/d-21phasiRNA loci (which would be >2,000 loci)?

We thank the reviewer for the comment. Our previous study identified 1,345 loci for *21PHAS*s, termed 21-nt phasiRNA clusters, using total small RNAs extracted from 0.5-mm anthers (Araki et al., 2022). We performed heatmap analysis of small RNAs interacting with AGO1b or AGO1d based on these 1,345 *21PHAS* loci. We have now described the *21PHAS*s used for the heatmap (page 7, line 24 – page 8, line 3).

We also performed a heatmap analysis with the anther small RNAs (see (a) in the figure below), and predominant patterns of small RNAs interacting with either AGO1b or AGO1d were also found in this analysis. Thus, we used the heatmap data focusing on AGO1b- and AGO1d-interacting small RNAs (see (b) in the figure below) in the revision as Fig. 3c.

3. Lines 11 to 13, page 3: the authors should clarify in which species (rice versus maize) these were observed.

As the reviewer requested, we have added “maize”, “Os”, and “rice” to clarify which species we were referring to (page 3, line 11-12).

4. Line 5, page 6: it’s not necessary to mention the obversion about SDS-PAGE, as the data are not shown.

We agree and have removed the statement about SDS-PAGE.

Reviewer #2 (Remarks to the Author):

Argonaute proteins (AGOs) are essential for the epigenetic regulation of plant development. AGO1 generally binds to miRNA to function in post-transcriptional gene silencing. The authors identified the function of two AGOs, AGO1b, and AGO1d, which interact with phased small interfering RNAs (phasiRNAs) to regulate rice anther development by acting as mobile carriers of the phasiRNAs from the somatic cell layers to the germ cells in anthers. They also performed three-dimensional (3D)-immunostaining using whole anthers, which to identify the specific subcellular localization of AGO1b/d in each somatic cell layer and germ cell. The authors emphasized a reproductive RNA silencing mechanism via the specific nuclear and cytoplasmic localization of AGO1b, AGO1d, and MEL1, in the anther phasiRNA pathway of rice pollen mother cells. However, the experiments could not support their conclusions, and more solid data are needed.

Major comments:

1. The author only performed the 3D multiple immunostaining to show the differing intracellular localization of AGO1b in the nucleus and AGO1d in the cytoplasm of all four cell types in the anther wall. However, The evidence provided in the paper is insufficient and needs to be confirmed by more methods. Such as confirmed by the immune colloidal gold technique.

In response to the reviewer's comments, first, we generated a guinea pig MEL1 antibody (Supplementary Fig. 5) to analyze the localization of AGO1b, AGO1d, and MEL1 at the same time with high resolution. We performed high-resolution multiple immunofluorescence with these three AGOs' antibodies and captured images using the Airyscan function of LSM 880 (Fig. 5), which enabled us to analyze samples with a resolution around half of the regular confocal system (i.e., 120 nm). These imaging results permitted spatial discrimination, including cellular and subcellular localization, between three AGOs, AGO1b, AGO1d, and MEL1 (Fig. 5a–h; page 9, line 14 – page 10, line 11). Moreover, we identified AGO1b-specific and AGO1d-specific foci in the cytoplasm of pollen mother cells (Fig. 5i–l; page 10, line 12-18).

We have also added negative-control imaging in this revision, in which samples were not treated with primary antibodies for AGO1b/d or MEL1, but were treated with secondary antibodies (Supplementary Fig. 6). Detection only of DAPI signals in the negative controls confirms that fluorescence from the immunostaining reflects the localization of the respective AGOs, and not

autofluorescence (page 10, line 1-6).

Thus, this high-resolution immunoinaging with negative controls (Fig. 5 and Supplementary Fig. 6) provides evidence for the specific cellular and subcellular localization of three AGOs in addition to the conclusions drawn from 3D immunoinaging (Fig. 4). We have added the imaging data and described them in the revision (Fig. 5; Supplementary Figs. 5, 6).

2. The author showed that AGO1b mRNA was expressed in the outer layers of the anther wall, and not in the inner tapetum layers or pollen mother cells. And AGO1d mRNA was also absent in the pollen mother cells by the immunostaining and RNA in situ hybridization assays, they are not enough to prove the conclusion. The quality of Fig.4 and Fig.5a-f is not good, and the RNA in situ hybridization signals cannot be clearly distinguished. Since it cannot be proved that AGO1b and AGO1d are only expressed in somatic cells, it cannot be said that they can be transferred to the mother cells.

On the other way, from the result of Tokunori Hobo et al., 2008 and Tang et al., 2010. (Tokunori Hobo et al., 2008, Various spatiotemporal expression profiles of anther-expressed genes in rice; Tang et al., 2010, Global gene profiling of laser-captured pollen mother cells indicates molecular pathways and gene subfamilies involved in rice meiosis. Plant Physiology), AGO1b, and AGO1d mRNA was expressed in the pollen mother cells. Therefore, more evidence is needed to confirm this result. Therefore, it could not be confirmed that the two genes translated into proteins and moved to the pollen mother cells.

We thank the reviewer for this comment and the microarray information. In response, we have added *in situ* hybridization data for *MEL1* mRNA localization as a PMC-specific marker (Fig. 6c), and we also captured clear images for *AGO1b*, *AGO1d*, and *MEL1* by adjusting the microscope to the most appropriate setting. We have also added *in situ* hybridization images of four anther locules and negative controls, to ensure the quality of the *in situ* hybridization (Supplementally Fig. 7). The contrasts in expression of these three *AGOs* between soma and germ (*AGO1b/d* in soma vs *MEL1* in germ) are similar to those of microarray data, as the reviewer mentioned (Hobo et al., PCP 2008; a modified version of their Supplementary Table 18 is attached below as additional data). Our qPCR data show vegetative tissue-specific *AGO1a* expression compared to reproductive tissue expression (Supplementary Fig. 9 in Araki *et al.*, 2020), while *AGO1a* is expressed in both microspore and tapetum, according to the microRNA data below. It is not clear whether the microRNA probes can

specifically detect each of the four *AGO1* subfamily genes, *AGO1a*, *AGO1b*, *AGO1c*, and *AGO1d*.

Table S18. Expression profiles of AGO genes (Modified Hobo et al., PCP 2008)

Relative expression level (Normalized data)							
Systematic Name	MEI-microspore-#1	MEI-microspore-#2	MEI-microspore-#3	MEI-microspore-#4	MEI-tapetum-#1	MEI-tapetum-#2	MEI-tapetum-#3
MEL1	11.239805	11.813073	10.7247095	11.242994	7.291567	7.631144	8.366665
AGO1b	8.527706	8.073689	7.8496604	7.8462257	10.006094	10.385461	10.284487
AGO1d	8.94669	8.168335	8.286039	7.419603	11.704018	12.116046	11.925235
AGO1a	9.977935	9.768103	9.426082	9.518162	10.6325445	10.76573	10.6614895

In general, it is very difficult to prove the complete absence of RNA even in mobile protein research. Studies of the mobile proteins florigen (FT/Hd3a), KNOTTED1 and SHR began with the identification of a difference in localization between mRNA and protein using *in situ* hybridization and promoter:GUS analysis, not using microarrays (Lucas et al., *Science* 1995; Nakajima et al. 2001; Tamaki et al., *Science* 2007).

In addition to the new *in situ* hybridization data, we performed qPCR using RNAs extracted from pollen mother cells in this revision. *MEL1* was highly expressed but we found extremely low expression of *AGO1b* and *AGO1d* (Fig. 6d; page 11, line 5-8). Therefore, we suggest that *AGO1b/d* are candidates for mobile proteins in anthers. However, we have avoided the word “absence” of *AGO1b/d* mRNAs in germ cells, and have mentioned but not over-asserted the mobile protein scenario in this revision.

2. The different localization of mRNA and protein of *AGO1b/d* subfamily genes in anther suggests that *AGO1b* and *AGO1d* migrate from outer layers to inner layers and pollen mother cells as carriers of phasiRNA. The authors did not separate the tissues from each layer of pollen mother cells for RNA –immunoprecipitation. It is not solid enough that the experimental evidence to support this conclusion.

As the reviewer commented, we have no data for tissue-specific RIPs. In the revised manuscript, as noted above, we have de-emphasized the descriptions of *AGO1b/d* as mobile proteins. Understanding the mobile functions of *AGO1b/d* is of great interest and will be the subject of future studies for us.

As we mentioned above, most mobile protein research started with the differential localization of mRNA and protein. Furthermore, the authors of a recent study proposed that *AGO1d* functions as a mobile protein via different mRNA/protein localization (Si et al., *Science China Life Sciences* 2022; Xiaofeng Cao’s group). We also published this manuscript as a preprint before the Cao group’s work

was published. By integrating our current data, recent phasiRNA/AGO studies, and the history of plant mobile protein research, it is not unreasonable to suggest that AGO1b/d are mobile proteins that act as phasiRNA carriers. In this revision, we have expanded the discussion of mobile proteins, citing the latest papers on AGO1d, florigen, KNOTTED1, and SHR (page 13, line 3, 4 and 8).

4. The experimental technology applied in the full text is relatively limited, actually, it does not clarify the significance and function of these two proteins to the AGO family or for the anther development in rice.

We agree with the reviewer, and for the revised manuscript, we performed high-throughput RNA sequencing using total RNAs extracted from 0.5-mm anthers of WT and *ago1b-1 ago1d-3* mutants (Supplementary Table 1). A total of 1,894 genes were upregulated in the *ago1b-1 ago1d-3* mutant, while 784 were downregulated, relative to their expression in WT plants (Supplementary Fig. 4a, b; false discovery rate (FDR) < 0.05, fold change (FC) > 5). Two up-regulated genes in *ago1b/d* double mutants, LOC_Os08g29669 and LOC_Os02g02870, are among nine phasiRNA target genes, whose mRNAs have been demonstrated to be cleaved by 21-nt phasiRNAs using meiocyte-based PARE/degradome sequencing¹³. There were also 111 upregulated *21PHASs*, which are precursors of 21-nt phasiRNAs, and 35 downregulated *21PHASs* in the *ago1b/d* double mutant (Supplementary Fig. 4c, d; FDR < 0.05, FC > 2). Based on these transcriptome data, we have raised the possibility of silencing via AGO1b/d due to the large numbers of upregulated genes/*21PHASs* (Supplementary Fig. 4; Supplementary Table 1; page 6, line 3-14).

Additionally, we further analyzed the anthers using 3D histological imaging methods for entire rice anthers, which can be used for distinguishing the internal structure of the anthers at the post-meiotic stage (Fig. 1h–o). Even though the *ago1b/d* double mutant has an outwardly normal anther structure, it showed abnormalities in the internal structures of the anther. The pollen grains were not filled in parts of the anther locule, some grains were crushed, and development of the inner layer was partially abnormal in 0.9-mm anthers (Fig. 1h–o). Thus, 3D imaging of the entire anther has enabled us to identify a variety of phenotypes even in the apparently normal anther locule of the double mutant. We have considered possible roles for AGO1b and AGO1d in the alignment, filling, and development of pollen grains during anther development in the revision (page 5, line 19 - page 6, line 2). These AGO1b/d germ functions in the normal anther may support the idea that they are mobile proteins, in

conjunction with the differences of AGO1b/d mRNA and protein localization.

Minor comments:

1. Figure 1d, The phenotypes of the six anthers of *ago1b-1 ago1d-3* are not completely consistent, what is the reason ?

We are also interested in the variable phenotypes of *ago1b/d* double mutants in anther development. Mutants of many factors in phasiRNA biogenesis (e.g., miR2118, MAGO, OsAGO1d, OsRDR6, and DCL5) show temperature- or photoperiod-dependent sterility, suggesting that phasiRNA biogenesis is involved in reproductive regulation via environmental responses, especially to temperature and day length.

The *ago1b/d* double mutants described in this manuscript were grown under conditions which resemble the natural summer conditions of northeastern Asia (long-day and 29.5–30 °C in the daytime), and revealed the variety of phenotypes in anther development. Thus, more severe phenotypes with complete defects might appear if the plants are grown under harsh environments, such as at low temperatures. Understanding the functions of AGO1b/d is therefore likely to be of interest in the related environmental fields and will be the subject of future studies.

2. In fact, the author did not do any experiments to prove that ARGONAUTE-mediated RNA silencing in anther development, so the title of this manuscript is not appropriate.

We have changed the title to “Spatial distribution of three ARGONAUTES regulates the anther phasiRNA pathway”.

Reviewer #1 (Remarks to the Author):

The authors have successfully addressed all of my concerns.

Reviewer #2 (Remarks to the Author):

Thanks to the authors for providing the additional experiments. Since this study highlights a new mode of reproductive RNA silencing via the specific nuclear and cytoplasmic localization of three AGOs, AGO1b, AGO1d, and MEL1, in rice pollen mother cells. More solid evidence is needed. Unfortunately, the present evidence even for mobile localization is slim.

Major comments :

- 1、 Though the author generated a guinea pig MEL1 antibody(Supplementary Fig. 5) to analyze the localization of AGO1b, AGO1d, and MEL1 at the same time with high resolution. The redone supplementary experiment cannot directly explain the mobile function of AGO1b and AGO1d. The analysis of AGO1b, AGO1d, and MEL1 localization by immune gold electron microscopy using anti-AGO1b, AGO1d, MEL1 antibodies in developing anthers can explain the mobile function of AGO1b and AGO1d more directly. It is suggested to supply this experiment.
- 2、 In situ hybridization experiment still needs to be improved, and the specific pollen stage and development stage can not be seen from the cell structure.
- 3、 Supplementary Figure 5: The anti-MEL1 antibody of the guinea pig is used in Wes analysis of anthers. The specificity of the antibody is unknown. It is unknown whether it can replace rice MEL1. There is no relevant evidence to support it. It is recommended to provide the relevant evidence

Minor comments :

- 1、 Figure 7C Each cell layer is not marked accordingly. It needs to be corrected
- 3、 Supplementary Figure 7. In situ hybridization of AGO1b, AGO1d, MEL1:
The pollen development stage in situ hybridization was not indicated. From the figure, a-b and c-d are not in the same stages, and the number of cell layers is different. Each cell layer is not marked accordingly. The in situ hybridization needs to be redone.

First of all, we would like to thank the reviewers for their constructive and valuable comments with regard to our manuscript. We have indicated (highlighted in yellow) the major changes made in the second revised manuscript and responded to Reviewer #2's comments below.

Additional results and major changes

- 1) New images for *in situ* hybridization, using a new probe to distinguish the four cell layers of anther walls (Fig. 6b; Supplementary Fig. 7c and d; Supplementary Table 10).
- 2) Validation of the guinea pig MEL1 antibody (Supplementary Table 9).

Reviewer #1 (Remarks to the Author):

The authors have successfully addressed all of my concerns.

Reviewer #2 (Remarks to the Author):

Thanks to the authors for providing the additional experiments. Since this study highlights a new mode of reproductive RNA silencing via the specific nuclear and cytoplasmic localization of three AGOs, AGO1b, AGO1d, and MEL1, in rice pollen mother cells. More solid evidence is needed. Unfortunately, the present evidence even for mobile localization is slim.

Major comments :

1、 Though the author generated a guinea pig MEL1 antibody(Supplementary Fig. 5) to analyze the localization of AGO1b, AGO1d, and MEL1 at the same time with high resolution. The redone supplementary experiment cannot directly explain the mobile function of AGO1b and AGO1d. The analysis of AGO1b, AGO1d, and MEL1 localization by immune gold electron microscopy using anti-AGO1b, AGO1d, MEL1 antibodies in developing anthers can explain the mobile function of AGO1b and AGO1d more directly. It is suggested to supply this experiment.

We thank Reviewer#2 for this suggestion. In this study, we indicated the cell-type difference of mRNA and protein localization between somatic anther wall (SOMA) and pollen mother cells (GERM) for the mobile AGO1b/d mechanism. Cell- and tissue-level dynamic change is a key factor in mobile studies, rather than organelle-level (i.e., subcellular) resolution. We would like to emphasize that 'mobile AGO1b/d' here refers to the movement of these proteins between cells/tissues,

and not to their movement between compartments within the same cell. Reviewer#2 suggests the use of immunogold electron microscopy at a high-resolution level instead, but we don't understand – and the reviewer doesn't explain – what could be observed in this system that would elucidate such an inter-cell mobile function. In the first revision, we have already added negative control data and new high-resolution data to reveal the cell-level differential localization of AGO1b/d as mobile proteins. Thus, we think there is little to be gained by using a higher-resolution level for illuminating mobile functions. To clarify mobile molecular functions, the behavior of regulators plays an essential role (Taoka *et al.*, 2011 *Nature*; Kitagawa *et al.*, 2022 *Science*). We are interested in understanding the mobile molecular functions of AGO1b/d and are now initiating studies that focus on candidate AGO1b/d regulators. We hope our subsequent research will contribute to a deeper understanding of mobile molecular function.

We have mentioned but not over-asserted “mobile function” in this revision, and discussed how our imaging and mutant analyses indicate the different functions between MEL1 and AGO1b/d in anther development (page 13, lines 7-12).

2、 In situ hybridization experiment still needs to be improved, and the specific pollen stage and development stage can not be seen from the cell structure.

As the reviewer requested, we performed *in situ* hybridization (ISH) for *AGO1d* localization using a new probe, which has been used in another publication (Fei *et al.*, 2016), and have replaced the images in Figure 6b and Supplementary Fig. 7c and d. The new ISH data enabled us to distinguish the four cell layers of the anther wall, in which the middle layers are thinner than the tapetum and pollen mother cells are less round than those after pachytene. The shape of pollen mother cells and anther wall layers indicate that the stage is between the premeiotic S-phase and the early meiosis.

The staging for pollen needs staining with DAPI or other markers of chromosome formation. However, since the ISH samples were embedded in paraffin with the whole inflorescence, it is unable to determine the stage. Judging from the structure of the anther wall and pollen mother cells, we have estimated the stage for ISH (Page 11, lines 2-5). The new *AGO1d* probe information has been added in Supplementary Table 10 and described, along with the citation, in this revision (Page 16 line 25 – Page 17 line 1; Page 30 lines 27-29).

3、 Supplementary Figure 5: The anti-MEL1 antibody of the guinea pig is used in Wes analysis of anthers. The specificity of the antibody is unknown. It is unknown whether it can replace rice MEL1. There is no relevant evidence to support it. It is recommended to provide the relevant evidence

We performed immunopurification (IP) on the guinea pig anti-MEL1 antibody (#1) and, moreover, a mass spectrometry analysis conducted by Kazusa DNA Research Institute confirmed that MEL1 was indeed immunopurified in MEL1-IP fractions, using this guinea pig antibody (Page 9 lines 21-23; Page 19 line 24 – Page 20 line 2). We have also shown the IP–mass spectrometry results in Supplementary Table 9. The guinea pig anti-MEL1 antibody (#1) was used for the immunostaining of Fig. 5

Minor comments :

1、 Figure 7C Each cell layer is not marked accordingly. It needs to be corrected

We did not realize that the reviewer was referring to Figure 6C instead of Supplementary Fig. 7C for “Figure 7C”. Each cell layer in the previous revision (Supplementary Fig. 7C) was correctly marked. Therefore, we have added the labels for each cell layer in Figure 6C in this second revision.

3、 Supplementary Figure 7. In situ hybridization of AGO1b, AGO1d, MEL1:

The pollen development stage in situ hybridization was not indicated. From the figure, a-b and c-d are not in the same stages, and the number of cell layers is different. Each cell layer is not marked accordingly. The in situ hybridization needs to be redone.

We re-performed ISH using a new probe as described above in our response to comment 2. The new ISH data enabled us to distinguish the four cell layers of the anther wall, in which the middle layers are thinner than the tapetum and pollen mother cells are less round than those after pachytene. The shape of pollen mother cells and anther wall layers indicate that the stage is between the premeiotic S-phase and early meiosis. We discussed the staging with Dr. Ono, one specialist for anther development, and added him to the Acknowledgements.

Moreover, we have marked each cell layer and pollen mother cells in Supplementary Fig. 7a, c, and e. We have indicated the sample stage in the figure legend of Supplementary Fig. 7.

The cell layers were not marked due to the difficulty we had in detecting signals in negative control figures using sense probes (b and d). However, the sample slides for the antisense and sense probes were derived from the same embedded anther, meaning that the same stage was used for the sense and antisense probes.

Reviewer #2 (Remarks to the Author):

The author has completed experiments to verify the specificity of antibodies, which can explain the relevant issues and addressed my concerns.

REVIEWERS' COMMENTS

Reviewer #2 (Remarks to the Author):

The author has completed experiments to verify the specificity of antibodies, which can explain the relevant issues and addressed my concerns.

We thank reviewer #2 for their valuable comment with regard to our revised manuscript.